# Effect of Different Activation Solutions and Protein Concentrations on Ide (*Leuciscus idus*) Sperm Motility Analysis with a CASA System

**DOI:** 10.3390/ani13040572

**Published:** 2023-02-06

**Authors:** Beata Irena Cejko, Sławomir Krejszeff, Agata Anna Cejko, Katarzyna Dryl

**Affiliations:** 1Department of Salmonid Research, Stanisław Sakowicz Inland Fisheries Institute, 10-719 Olsztyn, Poland; 2Department of Aquaculture, Stanisław Sakowicz Inland Fisheries Institute, 10-917 Olsztyn, Poland; 3Department of Fisheries Bioeconomics, Stanisław Sakowicz Inland Fisheries Institute, 10-719 Olsztyn, Poland; 4Department of Gamete and Embryo Biology, Institute of Animal Reproduction and Food Research, Polish Academy of Sciences, 10-243 Olsztyn, Poland

**Keywords:** ide, sperm motility, sperm adhesion, albumin, casein, CASA

## Abstract

**Simple Summary:**

Computer-assisted Sperm Analysis (CASA) systems are currently used to determine the sperm motility of various fish species. Our previous research indicated that one of the problems with CASA analysis is that sperm adheres to glass slides, which was noted in common carp (*Cyprinus carpio*). Supplementing activation solutions (AS) with proteins such as bovine serum albumin (BSA) or casein eliminates this problem and improves sperm motility. However, there is a lack of information regarding the minimal effective concentration of these proteins for ide (*Leuciscus idus*) sperm motility measurements. The present study is the first to compare four AS-Woynarovich, Lahnsteiner, Kucharczyk, and Perchec–for ide sperm motility activation. Woynarovich and Perchec solutions were selected and supplemented with different concentrations of BSA (0.25, 0.5, 1.0, and 2.0%) and casein (0.25, 0.5, 1.0, and 2.0%) for further motility measurements. Compared to pure Woynarovich and Perchec solutions (without proteins), the addition of the minimum concentration of BSA and casein (i.e., 0.25%) significantly improved CASA parameters of ide sperm. This dose was also sufficient for preventing sperm from adhering to glass slides. The different BSA (0.5–1.0%) and casein (1.0–2.0%) concentrations in Woynarovich and Perchec solutions affected CASA parameters differently.

**Abstract:**

The purpose of this study was to compare four activation solutions (AS)—Woynarovich, Lahnsteiner, Kucharczyk, and Perchec—with the addition of 0.5% bovine serum albumin (BSA) for ide (*Leuciscus idus*) sperm activation and analysis with a CASA system. It was found that ide sperm can be activated using each AS within a pH range of 7.4–9.0 and an osmolality range of 160–200 mOsm kg^−1^. The effect of Woynarovich and Perchec solutions supplemented with BSA and casein at concentrations of 0.25, 0.5, 1.0, and 2.0% were also analyzed during the experiment. These two AS without protein supplementation (pure solutions) were the controls. Woynarovich and Perchec solutions supplemented with the minimum BSA concentration (i.e., 0.25%) significantly improved sperm motility (89.05% and 86.63%, respectively) compared to the controls (20.39 and 28.48%, respectively). Similar increases were also noted in progressively motile sperm (PRG, %), the curvilinear velocity of sperm (VCL, µm s^−1^), and the amplitude of lateral head displacement (ALH, µm). A similar trend in CASA parameters was also noted when casein was added to Woynarovich and Perchec solutions at a concentration of 0.25%. We concluded that 0.25% doses of each of the proteins were sufficient to prevent sperm adhesion to glass slides, and they can be used in research on ide sperm motility measurements.

## 1. Introduction

Ide (*Leuciscus idus*) is a member of the family Leuciscidae and is widely distributed in Europe, especially in the lower reaches of rivers and lakes. This species is considered a valuable component of the trophic pyramid and a biomonitoring agent of water. Although ide is of little economic significance, it is important in recreational fisheries and in propagation as an ornamental fish. In controlled reproduction of other species belonging to the Leuciscidae family, ide was identified as a model species [1,2,3].

Several reports regarding domestication and artificial reproduction and their effects on ide egg quality have been published over the past fifteen years [4,5,6,7,8]. Studies on the characteristics of ide sperm and its storage in conditions of reduced metabolism have also been conducted. For example, hormonal treatment has been found to significantly impact the reproductive performance of males, including sperm quantity and quality [9]. Siddique et al. [10] described the composition of seminal plasma and the motility and kinetic parameters of ide sperm and pointed out that the seminal plasma had an approximate osmolality of 250 mOsm kg^−1^ and a pH of 8.0 and that Na^+^, K^+^ and Cl^−^ were the dominant ions in the reproductive system of this species. Our recent research also indicated the possibility of short-term ide sperm storage for two weeks, which may improve the reproduction of this species under hatchery conditions [11]. Similarly, Bernáth et al. [12] reported that ide sperm cryopreservation had no negative effects on motility, hatching rates, or larval malformation, which could efficiently support the gamete management of this species and its conservation. The results of these studies are promising, especially with regard to improving controlled ide reproduction and producing high-quality stocking material. Because ide spawn in their natural habitat, a limited number of spawners that produce gametes of moderate quality are usually obtained [9,10,13,14]. This is why appropriate methods of sperm collection and quality assessment are important stages of controlled ide reproduction and sperm used for fertilization, short-term storage, or cryopreservation. This is particularly important since ide production has increased over the past decade, especially in Eastern Europe [15].

Computer Assisted Sperm Analysis (CASA) systems are used widely to analyze fish sperm quality, and they are powerful, useful tools for scientific and ecological purposes [16,17]. CASA can also be applied in ide breeding programs since its results, such as motility (MOT, %) and sperm kinetic parameters, are markers of sperm quality and fertilization capacity [11,12]. However, CASA results are affected by a number of factors that must be considered during sperm analysis [18]. One of these is sperm adhering to glass slides, which is the main technical problem of CASA analysis.

Sperm adhering to glass slides results in low percentages of motile sperm and lower speeds that can produce inadequate data when CASA systems are used to measure sperm motility. For example, Siddique et al. [10] used fresh water to activate ide sperm movement, and motility (MOT) and sperm velocity (VCL) were 40% and 30 µm s^−1^, respectively. Such low parameters probably resulted from sperm adhering to glass slides, which reduced sperm motility values. Therefore, the addition of substances that reduce the adhesive properties of sperm is a necessary step in the preparation of activation solutions (AS) for sperm motility determinations. This is not yet well understood in fishes, and different proteins such as bovine serum albumin (BSA) or casein might be useful in limiting sperm adhesion to glass slides. Kowalski et al. [19] confirmed that supplementing Lahnsteiner buffer with 0.25% BSA and Perchec solution with 0.5% BSA increased sperm motility compared to pure solutions used for common carp (*Cyprinus carpio*) sperm activation. A minimal (i.e., 0.25%) casein concentration added to Perchec, and Lahnsteiner solutions can also be used to improve common carp sperm motility [19]. This indicates that the addition of BSA and casein to AS significantly affects the reliability of the results of CASA analysis.

Information regarding optimizing measurements of ide sperm motility using CASA is lacking. There is also no data on the minimal effective concentrations of BSA or casein for ide sperm motility measurements. This is why the aim of the present study was to compare the effects of the most commonly used Woynarovich, Lahnsteiner, Kucharczyk, and Perchec activation solutions supplemented with 0.5% BSA on ide sperm motility and kinetic parameters. In addition to testing 0.5% BSA, other concentrations of this protein (0.25%, 1.0% and 2.0%) and different concentrations of casein (0.25%, 0.5%, 1.0%, and 2.0%) added to Woynarovich and Perchec solutions were tested to determine the minimal effective concentrations of each protein when analyzing ide sperm with a CASA system.

## 2. Materials and Methods

### 2.1. Ethics Statement

The research was conducted with the permission of the Local Ethical Committee in Olsztyn, Poland, No. 30/2011, valid for the years 2011–2015.

### 2.2. Males Origin and Management

Ide males (n = 10) with an average body weight of 1250 g were collected in 2012 from the Łyna River downstream from Lake Mosąg (53°51′49.1″ N 20°23′35.2″ E) located in northern Poland. The fish were caught in March at a water temperature of 2–4 °C and transported to the hatchery of the Aquaculture and Ecological Engineering Center in Olsztyn. Next, they were maintained in 1000 L tanks in a RAS fitted with the instrumentation and equipment required to fully manipulate the environmental conditions of temperature and photoperiod [20]. After 10 days of acclimation at 5 °C, the water temperature was increased to 10 °C for five days. A constant 12L: 12D (light: dark) photoperiod was applied throughout the study.

### 2.3. Males Hormonal Treatment and Sperm Collection

After acclimatization, ide males were subjected to hormonal stimulation by intraperitoneal injection of a single dose of Ovopel (Unic-trade, Hungary) of 0.5 a pellet kg^−1^. One pellet of Ovopel typically contains 18–20 μg mammalian GnRH analog, i.e., D-Ala^6^Pro^9^NEt-mGnRH, and 8–10 mg of metoclopramide [21]. Sperm from ten males (n = 10) was collected manually according to the procedure described by Cejko et al. [9]. Prior to manipulation, the males were anesthetized using MS-222 (Sigma-Aldrich, St. Louis, MO, USA) at a dose of 150 ppm. The fish were prepared for sperm collection by drying their abdomens to avoid possible sperm contamination with urine, feces, or blood, and then a gentle abdominal massage was applied to them.

### 2.4. Sperm Motility Analysis Using Different Activation Solutions

Four AS with different ionic compositions, pH values, and osmolality were used to activate ide sperm (Table 1). Sperm from 5 males was used to conduct this experiment.

To prevent sperm samples from adhering to microscope slides, 0.5% bovine serum albumin (BSA; Sigma-Aldrich, St. Louis, MO, USA) was added to each AS. The percentage of motile sperm (MOT, %), progressively motile sperm (PRG, %), the curvilinear velocity of sperm (VCL, µm s^−1^), movement linearity (LIN, %), beat cross frequency (BCF, Hz), and the amplitude of lateral head displacement (ALH, µm) were determined using the CASA system. To measure motility and sperm kinetic parameters, a 1 μL ide sperm sample was activated in 100 μL of a selected AS, after which a 1 μL drop of the sperm and solution mixture was placed on a microscope slide (12 wells, 30-μm deep, Teflon-coated slide glass, Tekdon, Inc., Myakka City, FL, USA).

Recordings were made approximately 6 sec after the activation of motility using a Basler 202 K (Basler, Ahrensburg, Germany) digital camcorder integrated with an Olympus BX51 (Olympus, Tokyo, Japan) microscope (lens Plan FL N 20×/0.5 NH ph 1). The recording speed was 47 frames per second. The first 200 frames of each recording were then analyzed using the CASA system (CRISMAS, ImageHouse, Ltd., Birmingham, UK). The picture settings and parameters of analysis were as follows: AcquireTime delay: 0; ImageFields max 40; ImagesPerRecord 200; SpermTracks min: 10,000; ClassMethod: VAP; CombineLevel: 10; NoMoveLng: 5; TrackImmotileLevel: 5; TrackMotileLevel: 25; TrackProgressiveSTRLevel: 80; TrackTailsUse: True; and VelocityMax: 400. For each sample of sperm activated with different AS two replicates were made and then the average for each CASA parameter were estimated. During CASA analysis, sperm samples and AS were stored in an open Eppendorf tube at 4 °C using a thermomixer (ThermoMixer C, Eppendorf, Hamburg, Germany).

### 2.5. Sperm Motility Analysis Using Different BSA and Casein Concentrations

Separate ide sperm samples (n = 5) were used to determine the effects of different BSA concentrations (other than that of 0.5%), and different concentrations of casein (Sigma-Aldrich, St. Louis, MO, USA) added to the AS. Woynarovich and Perchec solutions were chosen and were supplemented with 0.25, 0.5, 1.0 and 2.0% BSA and 0.25, 0.5, 1.0 and 2.0% casein. The controls were pure Woynarovich (0.0% BSA or casein) and pure Perchec solutions (0.0% BSA or casein). CASA analyses were performed according to the procedures described above in Section 2.4.

### 2.6. Statistical Analysis

The results are presented as the mean ± SEM, and the differences were considered significant at *p* < 0.05. Prior to statistical analysis, percentage data were subjected to normalization using arc-sin transformation. Normality and lognormal tests were also performed before analysis, and the normal distribution of samples was checked. The significance of possible differences in the values of CASA parameters among the different AS used for ide sperm activation was determined with one-way analysis of variance (ANOVA) and Tukey’s post hoc test. The differences between BSA and casein supplementation of Woynarovich and Perchec solutions on CASA parameters was determined with a 2-way analysis of variance (ANOVA) and Tukey’s post hoc test. Differences between CASA parameters measured at the same protein concentrations in each solution were tested with the paired t-test. GraphPad Prism v.8 (GraphPad Software Inc., USA) was applied to perform the statistical analysis.

## 3. Results

### 3.1. Effect of Different Activation Solutions on CASA Parameters of Ide Sperm

The highest MOT parameter was determined with Woynarovich solution (71.54%), although the values of this parameter were not significantly different from those of Lahnsteiner (63.92%), Kucharczyk (61.29%), or Perchec (52.21%) solutions used for ide sperm activation (Figure 1a; *p* > 0.05). No significant differences were noted in the PRG parameter among the different AS used to activate ide sperm motility, the values of which for Woynarovich, Lahnsteiner, Kucharczyk, and Perchec solutions were 34.06%, 36.65%, 34.86%, and 27.93%, respectively (Figure 1b; *p* > 0.05). The highest VCL was noted with a Kucharczyk solution (118.90 µm s^−1^), and the values of this parameter differed significantly from those determined with a Perchec solution (94.50 µm s^−1^) (Figure 1c; *p* < 0.05). The were no differences for LIN and BCF among the AS used; 74.94% and 9.28 Hz, 79.43% and 9.18 Hz, 81.01% and 9.07 Hz, 77.05% and 9.43 Hz, respectively for Woynarovich, Lahnsteiner, Kucharczyk, and Perchec solutions (Figure 1d,e; *p* > 0.05). The highest ALH parameter was noted with a Woynarovich solution (0.98 µm), and the values of this parameter differed significantly from those determined with a Perchec solution (0.91 µm) (Figure 1f; *p* < 0.05). Based on these results, Woynarovich and Perchec solutions were used to determine the effects of different BSA and casein concentrations.

### 3.2. Effects of Pure Woynarovich and Perchec Solutions on CASA Parameters of Ide Sperm

Reduced values of CASA parameters MOT, PRG, VCL, and ALH were noted when pure Woynarovich (20.39%, 3.64%, 62.58 µm s^−1^ and 0.72 µm, respectively) and Perchec (28.48%, 8.68%, 83.84 µm s^−1^ and 0.81 µm, respectively) solutions were used to activate ide sperm motility (Figure 2a-c,f and Figure 3a–c,f). Moreover, significant differences were noted for MOT and PRG parameters between Woynarovich and Perchec solutions without the addition of proteins (Figure 2a,b and Figure 3a,b; *p* < 0.05). No differences were noted between pure Woynarovich and Perchec solutions for parameters VCL, LIN, BCF, or ALH (Figure 2c–f and Figure 3c–f; *p* > 0.05).

### 3.3. Effects of Woynarovich and Perchec Solutions Supplemented with Different BSA Concentrations on CASA Parameters of Ide Sperm

Table 2 presents detailed characteristics (mean ± SEM) of CASA parameters after ide sperm motility activation with Woynarovich and Perchec solutions supplemented with different BSA concentrations. Compared to pure Woynarovich and Perchec solutions, these solutions supplemented with BSA in concentrations of 0.25–2.0% resulted in significant increases in the MOT parameter of ide sperm, although there were no differences among the 0.25–2.0% BSA concentrations in motility parameter measurements. Significantly higher MOT parameters were noted when Perchec solution was supplemented with 0.5% (93.33%) and 1.0% (93.33%) BSA compared to Woynarovich solution supplemented with the same concentrations of this protein (88.35% and 88.35%), (Figure 2a; *p* < 0.05). There were no differences in PRG parameters between Woynarovich (range 35.67–43.72%) and Perchec (range 35.83–43.89%) solutions supplemented with some BSA concentrations (Figure 2b; *p* > 0.05). However, supplementing the Perchec solution with 0.5% BSA resulted in a significantly increased VCL parameter (144.74 µm s^−1^) compared to that of the Woynarovich (136.96 µm s^−1^) solution (Figure 2c; *p* < 0.01). Significant increases in the LIN parameter were noted when Woynarovich solution was supplemented with 2.0% (78.32%) compared to pure AS (67.00%), but there were no differences among Perchec solutions supplemented with different concentrations of BSA (Figure 2d; *p* < 0.05). Compared to the Perchec solution, the differences between the Woynarovich solution supplemented with 0.25% BSA compared to pure Woynarovich solution were significant for the BCF parameter (Figure 2e: *p* < 0.05). Woynarovich and Perchec solutions supplemented with BSA in concentrations of 0.25–2.0% significantly affected the ALH parameter compared to pure AS. Moreover, the highest ALH parameter (1.17 µm) was noted when the Perchec solution was supplemented with 1.0% BSA compared to the Woynarovich solution (1.09 µm) with the same concentration of protein (Figure 2f; *p* < 0.05).

### 3.4. Effects of Woynarovich and Perchec Solutions Supplemented with Different Casein Concentrations on CASA Parameters of Ide Sperm

Table 3 presents detailed characteristics (mean ± SEM) of CASA parameters after ide sperm motility activation with Woynarovich and Perchec solutions supplemented with different casein concentrations. Compared to pure Woynarovich and Perchec solutions, supplementing these solutions with casein at concentrations of 0.25–2.0% resulted in significant increases in the MOT parameter of ide sperm, although no differences were noted among the protein concentrations used (Figure 3a; *p* > 0.05). Similar results were observed for the PRG parameter; the values of this parameter were 35.10–40.37% with Woynarovich and 35.44–40.01% with Perchec solutions (Figure. 3b; *p* > 0.05). The VCL parameter noted with Perchec solution supplemented with 2.0% casein (148.18 µm s^−1^) was higher than that of Woynarovich solution (134.21 µm s^−1^) supplemented with the same protein concentration (Figure 3c; *p* < 0.01). A significant increase in the LIN parameter was noted when the Woynarovich solution was supplemented with 1.0% casein (76.76%) compared to the Perchec solution (72.76%) (Figure 3d; *p* < 0.01). No differences in the BCF parameter of ide sperm were noted among different casein concentrations with Woynarovich (8.18–8.65 Hz) or Perchec (7.89–8.75 Hz) solutions used for sperm activation (Figure 3e; *p* > 0.05). Compared to pure Woynarovich and Perchec solutions, supplementing them with casein in concentrations of 0.25–2.0% resulted in significant increases in the ALH parameter of ide sperm (Figure 3f; *p* > 0.05).

## 4. Discussion

The results of the present study show that ide sperm tends to adhere to glass slides during CASA analysis when it is activated with pure Woynarovich and Perchec solutions. But supplementing either solution with a minimum concentration of BSA (0.25%) significantly improved sperm motility (89.04% and 86.63%, respectively) compared to the controls (20.39 and 28.48%, respectively). Supplementing Perchec solution with 0.5–1.0% BSA resulted in higher percentages of motile sperm (MOT) and higher values of the curvilinear velocity of sperm (VCL) and the amplitude of lateral head displacement (ALH) compared to Woynarovich solution supplemented with the same BSA concentration. Casein also prevented sperm adhesion to glass slides, so Woynarovich and Perchec solutions supplemented with 0.25% casein can also be used to activate ide sperm motility for CASA analysis.

Optimizing measurements of sperm motility is a key step in sperm quality analysis. The composition of activation solutions (concentrations of Na, K, and Ca ions), osmolality, and pH influence the speed and dynamics of sperm movement, which, depending on the fish species, can last from a few seconds to a few hours [22]. Research has been conducted on ide sperm quality using different solutions for sperm motility activation. For example, we demonstrated previously that sperm motility in a colored form of European ide (*Leuciscus idus aberr orfus*) can be activated using Lahnsteiner solution, but sperm motility did not exceed 50% during the experiment [14], and when we used Kucharczyk solution to activate ide sperm, its motility did not exceed 40% [13]. Bernáth [12] used a Perchec solution to activate ide sperm motility; however, sperm velocity (VCL) was very low and did not exceed 60 µm s^−1^. Reduced sperm motility can be caused by urine contamination, which often occurs when males are stripped or when inadequate techniques (visual or CASA observations) or methods (AS supplemented with protein) are used during sperm motility measurements [23]. Based on our observations, the values of CASA parameters recorded after activation with the Woynarovich solution are similar to those observed following activation with Lahnsteiner, Kucharczyk, and Perchec solutions supplemented with 0.5% BSA. Significantly lower ALH parameters were noted with the Perchec solution compared to the Woynarovich solution. While there are no data regarding the importance of ALH parameters in ide reproduction, it has been confirmed that, in addition to MOT and VCL [24,25], ALH is also a bio-marker of sperm quality in Cypriniformes fishes such as crucian carp (*Carassius carassius*) [26,27] and barbel (*Barbus barbus*) [28]. It should also be noted that the Perchec solution contains a small amount of potassium ions (5 mM) and has a lower osmolality (160 mOsm kg^−1^) than the Woynarovich solution, which might have affected the sperm quality of this species following activation. A significant relationship between sperm motility and osmolality has also been noted in another Leuciscidae fish, bleak (*Alburnus alburnus*), in which low seminal fluid osmolality and weak motility was confirmed [29].

The problem of fish sperm adhering to glass slides has not yet been well investigated or explained. However, common carp sperm adhesion to glass slides has been reported [19]. In our research, we have found that ide sperm tends to adhere to glass slides, which impedes proper CASA analysis. Reduced MOT, PRG, and VCL values after sperm activation with pure AS compared to AS supplemented with 0.25–2.0% BSA were noted. Interestingly, we noted higher MOT and PRG values after sperm activation with pure Perchec solution (28.48 and 8.68%, respectively) than with pure Woynarovich solution (20.39 and 3.64%, respectively). While these differences were difficult to explain, Kowalski et al. [19] indicated that sperm adhesion might have been related to the ionic composition and osmolality of pure AS used during common carp motility measurements. These authors indicated that the lower osmolality of the Perchec solution (160 mOsm kg^−1^) resulted in lower sperm adhesion in contrast to that observed with the Lahnsteiner solution, which had a higher osmolality (200 mOsm kg^−1^).

A BSA concentration of 0.5% was the one most frequently applied with Kucharczyk [7], Lahnsteiner [14], and Woynarovich [11] solutions that was sufficient for performing ide sperm CASA analysis. On the other hand, 1.0% BSA was also used for ide sperm activation with Perchec solution [12]; however, the VCL values determined after sperm activation were very low (60 µm s^−1^) and close to those we noted following sperm activation with pure Woynarovich (62.58 µm s^−1^) and Perchec (83.84 µm s^−1^) solutions. In our present study, we found that a dose of 0.25% BSA significantly increased ide sperm motility and kinetic parameters. Significantly higher MOT, VCL, and ALH values were noted when Perchec solution supplemented with 0.5–1.0% BSA was used as compared to the Woynarovich solution. Common carp MOT, VCL, and ALH values were higher with the Perchec solution supplemented with 1.0–2.0% BSA than they were with the Lahnsteiner solution supplemented with the same concentrations of BSA [19]. This indicated that the Perchec solution could be used successfully as an AS for Cypriniformes fish such as common carp and ide. However, different BSA concentrations in AS significantly affected CASA parameters in each species.

Our results indicated that casein had a positive effect at a concentration of 0.25% on ide sperm motility (MOT, PRG, VCL, and ALH) in both Woynarovich and Perchec solutions. However, the increase in casein concentration from 0.5 to 2.0% in each of the AS had no significant impact on the CASA parameters of ide sperm. Similarly, the minimum dose of casein (0.25%) was added to Lahnsteiner and Perchec solutions, and this significantly improved common carp sperm motility [19]. Interestingly, regardless of the casein dose applied (0.25–2.0%), common carp sperm motility was significantly higher after sperm activation with the Perchec solution compared to the Lahnsteiner solution. Similar results were observed when Woynarovich and Perchec solutions were supplemented with casein in our present study. However, for casein, fewer differences in CASA parameters (VCL and LIN) between Woynarovich and Perchec solutions were noted. The results we obtained indicated that casein could be an alternative protein that can be added to an AS during ide sperm CASA analysis. Depending on the research project, it also permits selecting the appropriate protein for determining optimal CASA parameters in a given species.

Another important factor to be considered is the number of frames per second (FPS) used during fish sperm motility analysis. FPS directly influences the results since CASA tracks the head position of sperm movement [18]. In the current study, we used 47 FPS, while Bernáth et al. [12] used 60 FPS. Recent studies indicated that using >200 FPS is more appropriate for sperm with faster sperm swimming speeds, such as that of eel (*Anguilla anguilla*), Atlantic salmon (*Salmo salar*), and Siberian sturgeon (*Acipenser baerii*) [30]. Therefore, sperm motility analysis must not only include the appropriate AS supplemented with the appropriate concentration of protein, either BAS or casein, but also the appropriate CASA system settings, all of which are crucial for optimizing standard methods for assessing fish sperm quality. Future research on the quality of the ide sperm should also address this issue.

## 5. Conclusions

Supplementing Woynarovich and Perchec activation solutions with 0.25% BSA or casein to limit ide sperm adhesion to glass slides was sufficient to perform CASA analysis. Moreover, the Perchec solution supplemented with 0.5–1.0% BSA was a more suitable AS for ide sperm activation than was the Woynarovich solution supplemented with the same concentrations of BSA. The CASA analysis method described in the present paper can be applied in research on ide sperm quality and that of other species of the family Leuciscidae.

## Figures and Tables

**Figure 1 animals-13-00572-f001:**
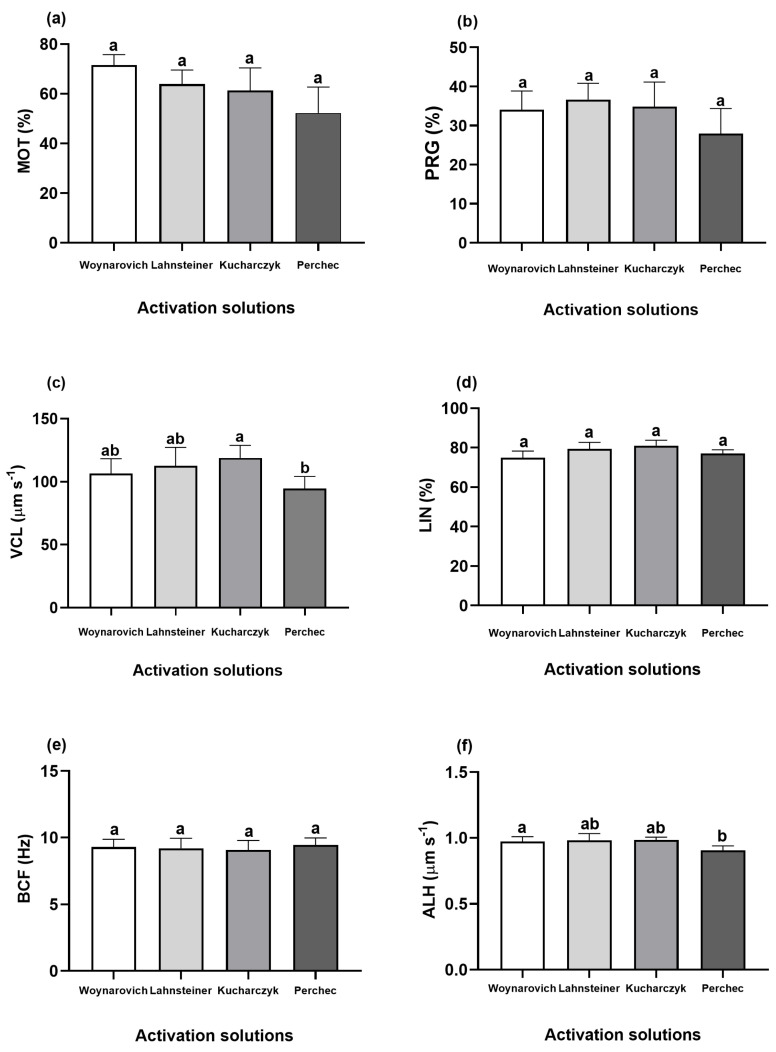
Percentages of motile sperm (**a**), progressively motile sperm (**b**), curvilinear velocity (**c**), movement linearity (**d**), beat cross frequency (**e**), and amplitude of lateral head displacement (**f**) in solutions supplemented with 0.5% BSA with different ion compositions, pH values, and osmolalties used to activate ide (*Leuciscus idus*) sperm (n = 5). Results presented as mean ± SEM (*p* < 0.05). Boxes labeled with different letters indicate statistically significant differences (*p* < 0.05).

**Figure 2 animals-13-00572-f002:**
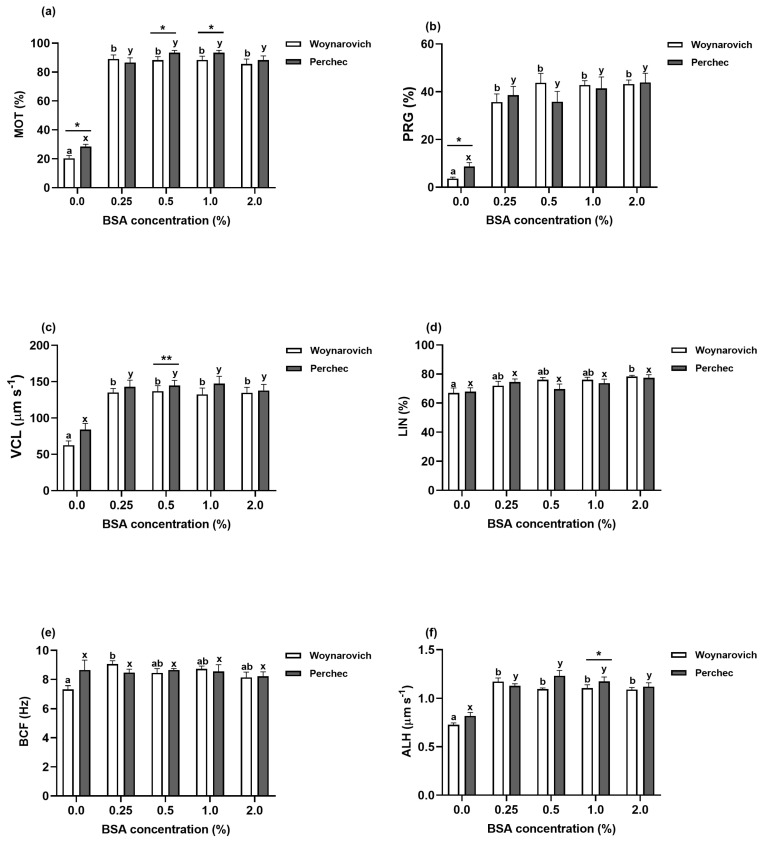
Percentages of motile sperm (**a**), progressively motile sperm (**b**), the curvilinear velocity of sperm (**c**), movement linearity (**d**), beat cross frequency (**e**), and amplitude of lateral head displacement (**f**) after activating ide (*Leuciscus idus*, n = 5) sperm motility with Woynarovich and Perchec solutions containing different BSA concentrations. Results presented as mean ± SEM (*p* < 0.05). Boxes marked with different letters (**a**,**b**) indicate significant differences among Woynarovich solutions supplemented with different BSA concentrations, and boxes marked with different letters (x,y) indicate significant differences among Perchec solutions supplemented with different BSA concentrations. Asterisks indicate the level of statistical significance of values measured in Woynarovich and Perchec solutions at the same protein concentrations (* *p* < 0.05; ** *p* < 0.01).

**Figure 3 animals-13-00572-f003:**
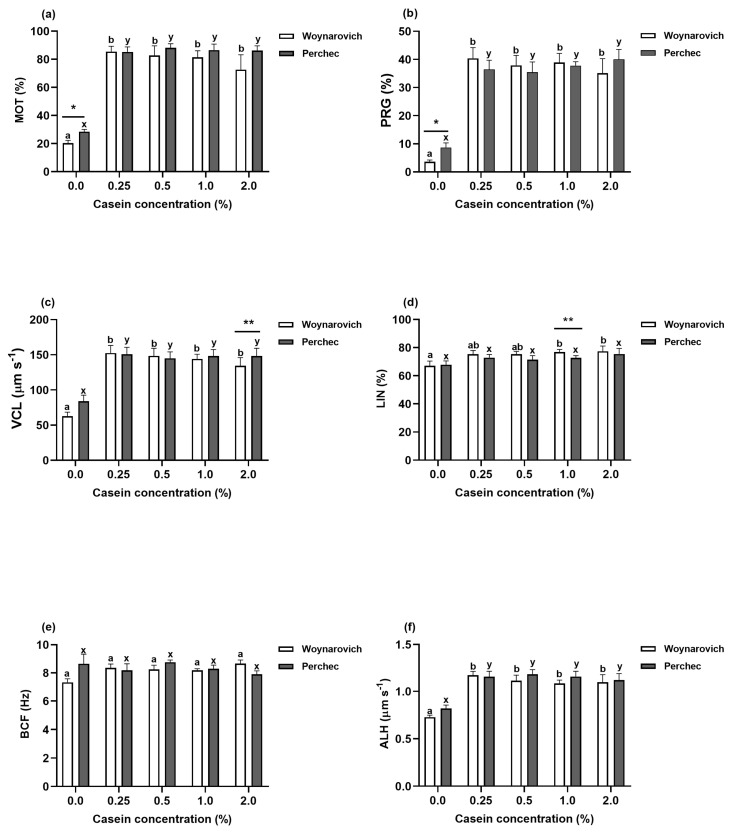
Percentages of motile sperm (**a**), progressively motile sperm (**b**), the curvilinear velocity of sperm (**c**), movement linearity (**d**), beat cross frequency (**e**), and amplitude of lateral head displacement (**f**) after activating ide (*Leuciscus idus,* n = 5) sperm motility with Woynarovich and Perchec solutions with different casein concentrations. Results presented as mean ± SEM (*p* < 0.05). Boxes marked with letters (**a**,**b**) indicate significant differences among Woynarovich solutions supplemented with different casein concentrations, and boxes marked with letters (x,y) indicate significant differences among Perchec solutions supplemented with different casein concentrations. Asterisks indicate the level of statistical significance of values measured in Woynarovich and Perchec solutions at the same protein concentrations (* *p* < 0.05; ** *p* < 0.01).

**Table 1 animals-13-00572-t001:** Activation solutions with different ionic compositions, pH values, and osmolality used for ide (*Leuciscus idus,* n = 5) sperm motility activation. * Each solution contained 0.5% bovine serum albumin (BSA).

Activation Solutions *	Solution Components	pH	Osmolality (mOsm kg^−1^)	References
Woynarovich	68 mM NaCl, 50 mM urea	7.7	180	[11]
Lahnsteiner	100 mM NaCl, 10 mM Tris	9.0	200	[14]
Kucharczyk	86 mM NaCl	7.4	167	[7]
Perchec	5 mM KCl, 45 mM NaCl, 30 mM Tris	8.0	160	[12]

**Table 2 animals-13-00572-t002:** Detailed characteristics (mean ± SEM) of CASA parameters after ide (*Leuciscus idus*, n = 5) sperm motility activation with Woynarovich and Perchec solutions supplemented with different BSA concentrations.

CASA Parameters	Activation Solutions									
	Woynarovich	Perchec	Woynarovich	Perchec	Woynarovich	Perchec	Woynarovich	Perchec	Woynarovich	Perchec
	BSA Concentration (%)									
	0.0		0.25		0.5		1.0		2.0	
MOT (%)	20.39 ± 1.89	28.48 ± 1.59	89.04 ± 2.80	86.63 ± 3.26	88.28 ± 2.42	93.33 ± 1.67	88.35 ± 2.67	93.33 ± 1.67	85.62 ± 3.40	88.28 ± 2.95
PRG (%)	3.64 ± 0.57	8.68 ± 1.63	35.67 ± 3.40	38.57 ± 3.65	43.72 ± 3.93	35.83 ± 4.26	42.78 ± 1.85	41.38 ± 4.77	43.21 ± 1.63	43.89 ± 3.83
VCL (µm s^−1^)	62.58 ± 5.80	83.84 8.69	135.07 ± 5.45	142.99 ± 9.01	136.96 ± 7.40	144.74 ± 6.96	132.50 ± 8.82	147.46 ± 9.90	134.68 ± 7.60	137.98 ± 8.23
LIN (%)	67.00 ± 3.47	67.87 ± 2.63	71.91 ± 3.06	74.50 ± 2.10	76.16 ± 1.58	69.65 ± 3.44	76.08 ± 1.74	73.68 ± 2.86	78.32 ± 0.78	77.38 ± 2.11
BCF (Hz)	7.32 ± 0.24	8.64 ± 0.68	9.05 ± 0.23	8.47 ± 0.22	8.44 ± 0.30	8.64 ± 0.10	8.72 ± 0.19	8.54 ± 0.46	8.13 ± 0.37	8.22 ± 0.29
ALH (µm)	0.72 ± 0.01	0.81 ± 0.03	1.17 ± 0.03	1.12 ± 0.01	1.09 ± 0.01	1.23 ± 0.05	1.10 ± 0.03	1.17 ± 0.04	1.09 ± 0.02	1.12 ± 0.03

**Descriptions:** MOT—percentage of motile sperm; PRG—progressively motile sperm; VCL—curvilinear velocity of sperm; LIN—movement linearity; BCF—beat cross frequency; ALH—amplitude of lateral head displacement.

**Table 3 animals-13-00572-t003:** Detailed characteristics (mean ± SEM) of CASA parameters after ide (*Leuciscus idus*, n = 5) sperm motility activation with Woynarovich and Perchec solutions supplemented with different casein concentrations.

CASA Parameters	Activation Solutions									
	Woynarovich	Perchec	Woynarovich	Perchec	Woynarovich	Perchec	Woynarovich	Perchec	Woynarovich	Perchec
	Casein Concentration (%)									
	0.0		0.25		0.5		1.0		2.0	
MOT (%)	20.39 ± 1.89	28.48 ± 1.59	85.47 ± 3.80	85.22 ± 3.53	82.67 ± 6.87	88.14 ± 2.87	81.39 ± 4.61	86.58 ± 4.20	72.62 ± 10.55	86.16 ± 3.39
PRG (%)	3.64 ± 0.57	8.68 ± 1.63	40.37 ± 3.80	36.43 ± 3.31	37.88 ± 3.54	35.44 ± 3.64	38.92 ± 3.23	37.76 ± 1.43	35.10 ± 5.16	40.01 ± 3.49
VCL (µm s^−1^)	62.58 ± 5.80	83.84 ± 8.69	152.41 ± 10.81	150.55 ± 9.95	148.4 ± 10.81	144.92 ± 9.12	144.09 ± 6.57	148.29 ± 9.09	134.21 ± 11.76	148.18 ± 11.00
LIN (%)	67.00 ± 3.47	67.87 ± 2.63	75.25 ± 2.68	72.84 ± 2.24	75.39 ± 1.92	71.41 ± 2.91	76.76 ± 1.91	72.76 ± 1.50	77.34 ± 3.75	75.47 ± 3.96
BCF (Hz)	7.32 ± 0.24	8.64 ± 0.68	8.35 ± 0.27	8.17 ± 0.47	8.25 ± 0.29	8.75 ± 0.15	8.18 ± 0.10	8.29 ± 0.25	8.65 ± 0.25	7.89 ± 0.24
ALH (µm)	0.72 ± 0.01	0.81 ± 0.03	1.17 ± 0.04	1.15 ± 0.05	1.11 ± 0.05	1.18 ± 0.04	1.08 ± 0.03	1.15 ± 0.05	1.09 ± 0.08	1.12 ± 0.06

**Descriptions:** MOT—percentage of motile sperm; PRG—progressively motile sperm; VCL—curvilinear velocity of sperm; LIN—movement linearity; BCF—beat cross frequency; ALH—amplitude of lateral head displacement.

## Data Availability

The data presented in this study are available on request from the corresponding author. The data are not publicly available for privacy reasons.

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
