# Peer review of "Effect of Different Activation Solutions and Protein Concentrations on Ide (Leuciscus idus) Sperm Motility Analysis with a CASA System"

_animals, 2023, doi:10.3390/ani13040572_

Round 1

Reviewer 1 Report

This study aims to improve the sperm quality analysis of Ide sperm by reducing sperm adhesion to slide during motility (by CASA) analysis. This seems to be an important study, however clarification surrounding what past studies have already been conducted in this area are needed to justify why this study was completed. E.g. previous studies on this topic in ide have already been published by the authors (according to this manuscript). Throughout the writing style needs attention. Some sentences need rewording for clarity. Many sentences start with “it was confirmed”, “that is why”, etc. with no context for the sentence.

Abstract: No justification for the study is included and no specific results. It would give more context to the reader to provide some numerical results (e.g. motility percentage) for the major findings.

Introduction: I find the introduction a little disordered with many ideas within paragraphs but not clear links between the ideas. E.g. In first paragraph there is a list of research uses for Ide followed by the statement that there have been reproductive studies done in recent years. Is the increase in reproductive studies (such as sperm storage) due to a need for captive breeding, or a reduced population in the wild, or does sperm quality have some sort of bearing on other studies mentioned (e.g. toxicology studies)?

L47-51: The wording of this sentence seems to be almost word for word the same as in another recent paper by the authors. See: Cejko, Beata Irena, et al. "Ide (Leuciscus idus) sperm short-term storage: Effects of different extenders and dilution ratios." Animal Reproduction Science (2022): 107155.

L54-57: Needs rewording for clarity. I also am not sure what the relevance of this information of female reproduction is to the current study on sperm quality.

L59: “reduced quality” by what measure and compared to what?

L65 What is meant by “optimization of reproductive biotechnology of ide”? can you be more specific.

L78: You state that CASA is “free from human error”, however I’m not sure I agree with this as the settings/parameters used with CASA can be subjective and change the output results significantly.

L81-82: It was confirmed by who and in what study? If from the present study this is not appropriate for the introduction.

L82-90: Needs rewording for clarity. This content also seems more relevant for comparison to present study within the discussion. I also question why the present study is needed if the study by the authors has already been published in a sub species of the species used in this study.

L91-93: It is not clear whether this is referring to the present study, another study or what different methods of sperm qualities assessments are even being referred to?

L98-99: have there been studies and/or analysis that confirm sperm are stuck to the slide as opposed to just low in motility?

L109-110: Why would you not want antioxidants? Can you explain this statement further.

Methods:

L134: Please provide year of capture (especially since ethics statement is effective only between 2011 and 2015) and GPS coordinates of capture.

L137: How many fish per 1000 L tank?

L149: Here is says n=5, but on L133 you say n=10. Please clarify which is correct.

L145-149: Can you provide an average dose per animal based on information provided and weights.

Table 1 needs units for osmolality; Activation solutions are labelled AS1, 2, 3, 4 but throughout paper this naming system is not used. Please change so consistent.

Statistical analysis: There are several papers showing that use of arcsine transformation is inappropriate. I think I more robust analysis would be to run generalized linear models (or similar) analysis.

Results: I think a summary table of each parameter would be beneficial. I also don’t understand why the controls are compared to each other, then all BSA treatments are compared to each other and then all casein are compared to each other individually. Should all treatments be compared to each other?

L234: Reduced values compared to what? Be specific.

L248: In the abstract you say 0.25-1.0% is optimal but here you say 0.25-2.0%. Please make consistent.

Discussion:

L317: Can you be more specific than “simple solution”.

L327-330: The link between this information and the subject of the paper (sperm sticking to slides) is unclear. Make the relevance clearer.

L330-331: Then why is the current study novel or worthwhile, if this has already been done in the same species.

L335: Why is the information on pre-incubation of eggs relevant to this paper? Please elaborate or delete as appropriate.

L341-343: This statement is confusing to me; since this paper doesn’t investigate ALH as a marker, there either is or isn’t previous data on this, but the current sentence is contradictory. Please revise for clarity and make clear whether data is from this, or a previous paper.

L351: You say BSA is traditionally added to AS but there are no supporting citations. Please add.

L357-359: I don’t think you’re results specifically look at osmolality. There are four activating solutions of different osmolality, however the composition of these are different so you are not looking at osmolality in isolation. To look at the effect of osmolality you would need to vary the osmolality of each of the AS individually.

L363: Can you report all BSA concentrations in the same units so that the information is easily compared.

L385: “casein may be added to AS as a “second-choice”” This has been written several times and is very repetitive. Additionally, since you didn’t really compare BSA to casein with your statistical analysis it’s unclear how this compares to the BSA results.

L405. You use the word “buffer” here when previously you say AS, activating solution. Use consistent terminology.

Author Response

Dear Reviewer 1

We have included our answers to the Reviewer 's questions in World file

Reviewer 2 Report

animals-2049434-peer-review-v2

Manuscript title:

"Efect of different activation solutions and protein concentration on ide (Leuciscus idus) sperm motility analysis using CASA system"

Review:

General comments:

The subject of the article is very important for practical use at spermatozoa motility detection  but the abstract and results should be rewritten. The main conclusion of the study was to indicate minimal effective concentration of alumin or  casein concetration to ide (Leuciscus idus) sperm motility measurements. The concentration of  0,25 % (it is pitty that lower concentration was not tested) is enough addition to eliminate spermatozoa stickiness (in 0,25-2% concentrations there are not significant differences in motility parameters).  

Moreover although some significant differences between 0,5 -1% BSA and casein of  Woynarovich and Perchec solutions were shown it seem to be rater random resulted from data distribution (n=5 and twice repetitions can be a few repetition), they seem to fluctuate so omit the data in the Abstract. Example: when compare CASA results in Woynarovich and Perchec solutions supplemented with  0.5% BSA in Figures 1 and 2 the tendency were not uniform. The Perchec solution was the worst solution in the first step of the study while favorite in the second step. The results of these two steps of the study are a bit confusing. [Consider to publish these results in two different papers or conduct additional tests to make tendency consistent in the first or second steps].

Detailed comments

Title

Change Efect to Effect

Citation patten

Unify the authors' names and title of manuscript  in the main body and citation pattern

L29 List the four activation solution tested in the study (possibly this part will be removed from the article)

L 36  first introduce the minimum concentration of BSA and casein that significantly improve motility reactively to pure Woynarovich and Perchec solutions.

L 36-38 Although the significant differences between Woynarovich and Perchec solutions was shown it seem to be random resulted cause from data distribution (n=5 and twice repetitions),  so omit the data in abstract.

L 40   0,25% is sufficient

L 185 what was the temperature of activating solution and microscope table?

L 221 change unit to µm s-1

L 226 the unit for ALH is µm, change for µm

L 249 there are not significant differences between 0,25 - 2% BSA addition  in motility parameters measurements  

L256 with same supplementation of BSA

L 285 no true, there are differences in Fig 3f

L 293-294 correct the sentence

L293 solutions supplemented with

L317-320 rewrite the sentence (0.5 %BSA is also mentioned in the first sentence)

L 321 compare to pure Perchec then to Woynarovich

L 387 delete “optimal”

L 404-410 rewrite

Recommendation for next study: Lower concentration of proteins should be tested to indicate minimal effective concentration for sperm motility measurements.

Fig 1 rewrite - after activation in solution of different ions….  with addition

Author Response

Dear Reviewer 2

We have included our answers to the reviewer's questions in World file

Reviewer 3 Report

Animals 2049434 V2

“Efect of different activation solutions and protein concentration on ide (Leuciscus idus) sperm motility analysis using CASA system”

In the present work, authors analysed the effect of 4 different solution supplemented with 0.5% BSA in the motility parameters analysed by CASA system evaluation.

The study is interest to Animals readers, as the topic is actual and original. The work was adequately performed and the results are reliable. The text is clear and objective and conclusions are consistent with results obtained.  Figures add important information.

I suggest minor changes, as some English revision of the manuscript.

-Please explain better the problem of sperm adhesion to glass surface in this and other species and its negative consequences for sperm evaluation.

-L 15 and L 16: improve the phrase construction

-Also correct in the Title to “Effect”

L 160: clarify better the solution that received, beside BSA, supplementation with casein.

-L 170: 1 µL of ide sperm corresponds to which sperm number? Was concentration measured by CASA system? Please revise and correct accordingly.

-L 171: again in the incubation, how long did incubation occur?

-Authors intend to evaluate some membrane integrity parameters, besides CASA evaluation?

-L 210: if results with one solution are only numerically superior, it is not strictly necessary to refer them. Only emphasize when significant differences really exist.

Author Response

Dear Reviewer 3

We have included our answers to the reviewer's questions in World file

Round 2

Reviewer 1 Report

The readability of this manuscript has been greatly improved and most changed have been addressed appropriately. 

Regarding sample size in methods: I presume experiment one also had a n=5, to make up the total n=10 indicated at the start of methods. Can this be added for clarity.

Regarding Statistical methods: I am aware the historical literature uses arcsin transformations, as you have indicated, and that many people still use it. I would draw your attention to these recent papers as an example of why statisticians no longer recommend using it:

https://esajournals.onlinelibrary.wiley.com/doi/10.1890/10-0340.1

https://onlinelibrary.wiley.com/doi/full/10.1002/hsr2.178

Author Response

Response to Reviewer 1 Comments

We would like to thank the Reviewer for her/his valuable comments and suggestions. We took them into account in the process of reviewing our work and made the required changes. We also hope that our answers to the all questions asked are clear and comprehensive. The MS has also been checked in terms of language, the final version is the Clean version.

The readability of this manuscript has been greatly improved and most changed have been addressed appropriately.

Regarding sample size in methods: I presume experiment one also had a n=5, to make up the total n=10 indicated at the start of methods. Can this be added for clarity.

Yes, the total amount of fish used in the experiment is n = 10. In the 1st experiment we used n = 5 and in the next 2nd experiment n = 5. This information is provided in the MS (Page 7, lines 227-228).

Regarding Statistical methods: I am aware the historical literature uses arcsin transformations, as you have indicated, and that many people still use it. I would draw your attention to these recent papers as an example of why statisticians no longer recommend using it:

https://esajournals.onlinelibrary.wiley.com/doi/10.1890/10-0340.1

https://onlinelibrary.wiley.com/doi/full/10.1002/hsr2.178

Thank you for the Reviewer's attention. We will take it into account in future studies in order to avoid potential errors related to statistical analysis. Each such comment is very valuable to us and improves our scientific workshop.
